# A method for campus-wide SARS-CoV-2 surveillance at a large public university

Terren Chang[1,2,3☯], Jolene M. Draper[1,3,4,5☯], Anouk Van den Bout[1,3,4,5☯], Ellen Kephart[1,3,4,5☯], Hannah Maul-Newby[1,3,4☯], Yvonne Vasquez[1,3,4,5☯], Jason Woodbury[6], Savanna Randi[1,5], Martina Pedersen[1,5], Maeve Nave[1,5], Scott La[1,5], Natalie Gallagher[1,5], Molly M. McCabe[1,4,5], Namrita Dhillon[1,4,5], Isabel Bjork[1,5], Michael Luttrell[7], Frank Dang[8], John B. MacMillan[1,2], Ralph Green[8], Elizabeth Miller[9], Auston M. Kilpatrick[10], Olena Vaske[1,2,3,4,5], Michael D. Stone[1,2,3]*, Jeremy R. Sanford[1,3,4,5]*

1 Colligan Clinical Diagnostic Laboratory, University of California Santa Cruz, Santa Cruz, California, United States of America, 2 Department of Chemistry and Biochemistry, University of California Santa Cruz, Santa Cruz, California, United States of America, 3 UC Santa Cruz Center for Molecular Biology of RNA, University of California Santa Cruz, Santa Cruz, California, United States of America, 4 Department of Molecular, Cellular and Developmental Biology, University of California Santa Cruz, Santa Cruz, California, United States of America, 5 UC Santa Cruz Genomics Institute, University of California Santa Cruz, Santa Cruz, California, United States of America, 6 Thirdwave Analytics, San Francisco, California, United States of America, 7 Business and Accounting Services, University of California Santa Cruz, Santa Cruz, California, United States of America, 8 University of California Davis, Davis, California, United States of America, 9 Student Health Services, University of California Santa Cruz, Santa Cruz, California, United States of America, 10 Department of Ecology and Evolutionary Biology, University of California Santa Cruz, Santa Cruz, California, United States of America

☯ These authors contributed equally to this work.
* jsanfor2@ucsc.edu (JRS); MDS@ucsc.edu (MDS)

**Data Availability Statement:** All relevant data are within the paper and its S1 Data files.

**Funding:** The work in this manuscript was supported by grants from the National Institutes of

## Abstract

The systematic screening of asymptomatic and pre-symptomatic individuals is a powerful tool for controlling community transmission of infectious disease on college campuses. Faced with a paucity of testing in the beginning of the COVID-19 pandemic, many universities developed molecular diagnostic laboratories focused on SARS-CoV-2 diagnostic testing on campus and in their broader communities. We established the UC Santa Cruz Molecular Diagnostic Lab in early April 2020 and began testing clinical samples just five weeks later. Using a clinically-validated laboratory developed test (LDT) that avoided supply chain constraints, an automated sample pooling and processing workflow, and a custom laboratory information management system (LIMS), we expanded testing from a handful of clinical samples per day to thousands per day with the testing capacity to screen our entire campus population twice per week. In this report we describe the technical, logistical, and regulatory processes that enabled our pop-up lab to scale testing and reporting capacity to thousands of tests per day.

## Introduction

The SARS-CoV-2 pandemic radically altered society, the economy, and global health. The rapid spread of SARS-CoV-2 is driven by a combination of asymptomatic carriers and pre-

Health to MDS (R01GM095850) and JRS (R35GM130361), as well as support from the University of California Santa Cruz and the County of Santa Cruz. Grants from the NIH and funding from the University of California Santa Cruz supported salaries of all authors except JW who is an employee of Thirdwave Analytics. The University of California Santa Cruz and the County of Santa Cruz supported instrumentation, information technology and build out of the lab. The Colligan family supported renovation of laboratory space. The funders had no role in study design, data collection and analysis, decision to publish, or preparation of the manuscript.

**Competing interests:** MDS and JRS are paid consultants and have an ownership interest in SummerBio, a commercial COVID testing laboratory that specializes in institutional surveillance. JW is an employee of Thirdwave Analytics. This does not alter our adherence to PLOS ONE policies on sharing data and materials.

symptomatic transmission [1, 2]. Diagnostic testing is one of the critical tools for breaking viral transmission chains [3, 4]. The combination of testing for symptomatic cases, tracing and testing of close contacts, and isolation of infected individuals proved to be a highly effective approach to control community spread of SARS-CoV-2 [5–7]. Unfortunately, testing capacity in the USA was constrained by supply chain shortages [8–11], paucity of commercial and medical center diagnostic labs with available bandwidth to scale testing rapidly, and an uncoordinated response at federal, state, and local levels [12–14].

Institutions of higher education were dramatically impacted by the pandemic. Instruction shifted to distance learning, research activity was curtailed, student activities and organizations ceased operations, and on campus housing was dramatically reduced [15, 16]. For college campuses, molecular diagnostic testing for asymptomatic or pre-symptomatic SARS-CoV-2 infections plays an important role in preventing community spread by identifying and isolating cases at the earliest stages of infection [17]. Such rapid turnaround typically requires an in-house testing facility. However, many college campuses, especially those without a medical school, lacked their own clinical diagnostic labs at the beginning of the pandemic, hindering the campus testing process and response. To address this challenge, the State of California developed a regulatory mechanism for creating "pop-up" laboratories by which an existing clinical laboratory could extend its license onto research space to allow for the increase in clinical testing capacity in the areas of need. The primary challenges for campus diagnostic labs are throughput, rapid turnaround time for test results, and a sampling frequency that is less than the viral incubation period [3]. Additional considerations such as the biospecimen type and the testing platform play important roles in successful implementation of an institutional-level testing program. The other critical component is a laboratory information management system (LIMS) capable of accessioning, tracing, and reporting in a timely manner to state and local public health agencies on thousands of samples per day through an automated, hands-free process. Finally, given the reality of limited budgets for testing, successful scaling requires that the test be as cost-effective as possible.

In response to the pandemic many university research laboratories pivoted to SARS-CoV-2 molecular diagnostic testing [18–22]. This was possible through a relaxed regulatory framework that enabled the creation of temporary COVID-19 testing sites operating under existing campus clinical laboratory licenses. These pop-up diagnostic laboratories capitalized on the ingenuity and expertise of faculty, students, and staff to develop and validate laboratory developed tests (LDTs, tests whose application is restricted to the given laboratory [23]) to serve the needs of their campus and local communities. In response to the early stages of the COVID-19 pandemic, we established a temporary SARS-CoV-2 testing site at the University of California Santa Cruz, in accordance with the Clinical Laboratory Improvement Amendments (CLIA'88, 42 USC § 263a) and the California Department of Public Health guidelines. Our clinical laboratory performs diagnostic testing of symptomatic patients for our community's safety-net healthcare providers and the UC Santa Cruz Student Health Clinic as well as asymptomatic screening on campus.

This paper builds upon the blueprint laid out by our colleagues at UC Berkeley [22], but describes the strategic choices we made to scale testing capacity and overcome the common barriers to SARS-CoV-2 diagnostic testing (Fig 1). In particular we adopted a distinct liquid handling strategy, using 96 well pipetting heads. By integrating this parallel sample processing strategy with our Laboratory Information Management System, we created a sample collection process that was compatible with 96 well format RNA extraction and facilitated paperless test requisition, accessioning and reporting. The liquid handling strategy also enabled a 'linear sample pooling' approach that allowed for efficient 10:1 pooling and rapid pool deconvolution, if necessary. Taken together, our strategy enabled us to scale cost-effective, rapid turnaround

**1 Sample Collection** into Micronic 1D/2D barcoded tubes from self-administered AN swab. 1D bar code used at testing kiosk to track tube.

**2 Sample Accessioning** racks of 96 sample tubes brought into testing lab and scanned on Ziath flatbed 2D barcode scanner, followed by upload of samples data into custom Laboratory Information Management System (LIMS).

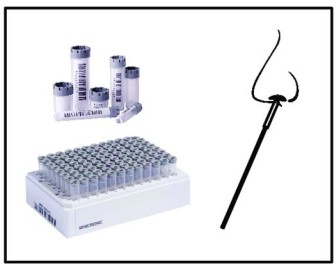

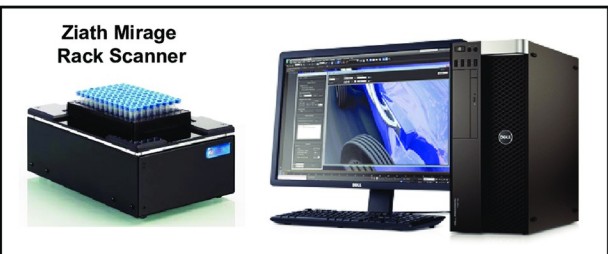

Ziath Mirage Rack Scanner

**3 Automated Sample and Liquid Handling.** Racks of 96 tubes are simultaneously decapped in Micronic CS700 capper/decapper. Liquid dispensing to prepare plates and samples for automated RNA extraction protocol perfomed with MultiFlo BioTek Dispenser, Formulatrix Mantis Dispenser, and Eppendorf epMotion pipetting robot.

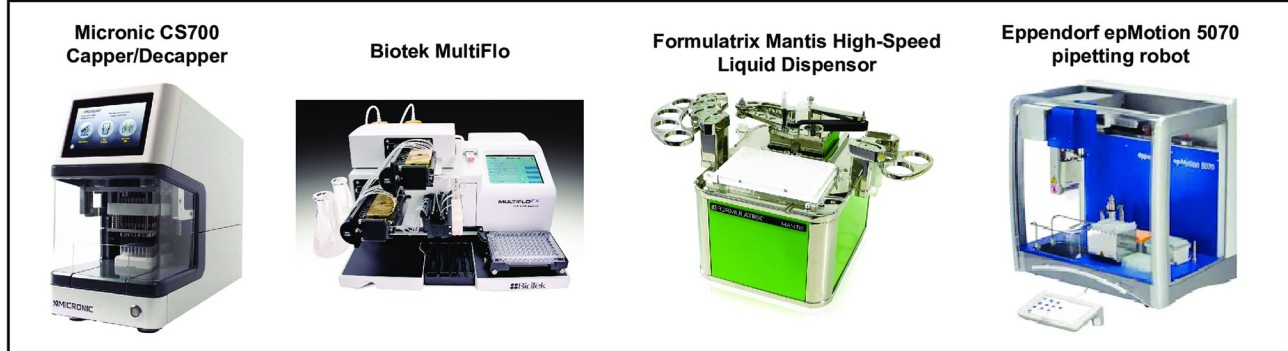

Micronic CS700 Capper/Decapper

Biotek MultiFlo

Formulatrix Mantis High-Speed Liquid Dispensor

Eppendorf epMotion 5070 pipetting robot

**4 Automated Sample Pooling and RNA Extraction.** Individual sample racks are pooled, followed by optimized automated RNA extraction on Agilent Bravo NGS liquid handling robot.

**5 PCR plate setup and run.** Four 96-well RNA storage plates are combined onto one 384-well RT-qPCR plate on the Bravo NGS, and run in the ABI QuantStudio 6 Pro.

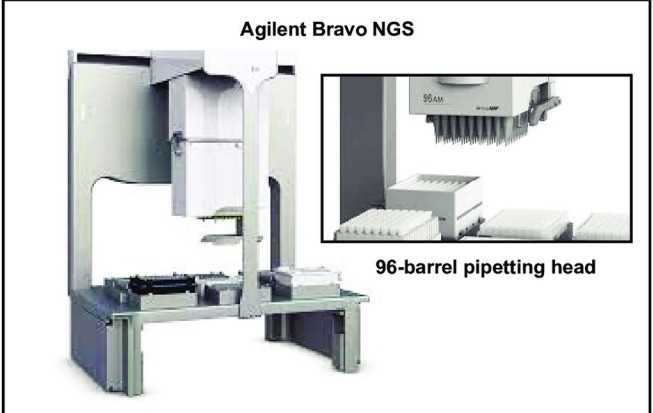

Agilent Bravo NGS

96-barrel pipetting head

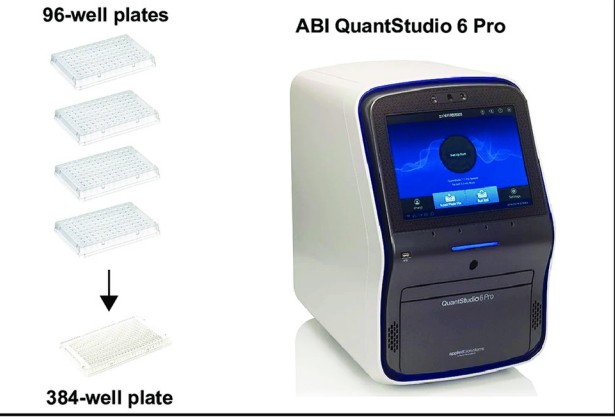

96-well plates

ABI QuantStudio 6 Pro

384-well plate

**Fig 1. Overview of the UC Santa Cruz Colligan clinical diagnostic laboratory workflow.**

testing capacity to thousands of tests per day, with a lean staff and a modestly equipped laboratory. We believe the approach outlined here can be widely implemented and will be useful for public health efforts during the current pandemic and future outbreaks of infectious disease.

## Methods

### Assembly of sample collection kits

Sample collection kits consist of racks of barcoded 1.4 mL tubes (Micronics) filled with 0.6 mL transport media (DNA/RNA Shield, Zymo Research). Following automated decapping (Micronics) of the clean tube racks, transport media is added using a MultifloFX liquid dispenser (BioTek) and recapped. The barcodes on each rack of tubes are scanned using a Zianth flatbed scanner and tube barcodes are uploaded into the LIMS. Racks containing collection tubes along with anterior nasal swabs (Typenex Medical) and additional caps are distributed to testing kiosks.

### Sample collection at campus kiosks

Participants in the campus asymptomatic testing program check in at a central desk using their student IDs or state issued IDs. After confirming their ID, the receptionist enables the student to select a sample collection tube and scan the 1D barcode. The barcoded tube is linked to an electronic test requisition form (eTRF) for the specific participant. The participant then brings the tube to a supervised self-collection site and swabs each nostril for 15 seconds, then dunks the swab in the DNA/RNA Shield transport media for 30 seconds, wipes the rim of the tube with a kimwipe and applies a clean cap. Sample tubes are then placed into a collection tube rack for transport to the laboratory.

### Automated RNA extraction and RT-qPCR

Each sample tube contains approximately 400 uL of nasal swab in DNA/RNA shield, extraction is initiated by adding 800 uL Viral RNA Buffer (VRB, ZymoResearch) to each sample using a MultifloFX liquid dispenser (BioTek). The addition of the Viral RNA Buffer plus reducing agent at this step has robust mucolytic properties which significantly reduces pipette tip clogging due to sample viscosity. Samples are titurated on the Bravo deck and 540 uL is transferred to a 1.2 mL deep well plate. RNA is extracted using magnetic beads (ZymoResearch) and eluted in 35 uL ddH2O and 5 uL is used as template for RT-qPCR. All automation scripts can be found here on github https://github.com/UCSC-CCDL/Bravo-protocol-files.

### LIMS, sample accessioning, and reporting

Custom laboratory information management system (LIMS) was developed in collaboration with Third Wave Analytics (San Francisco) using the Salesforce Lightning Platform and Experience Cloud. This system tracks every sample, well position, plate, RT-qPCR result and applies logic tables to call the presence or absence of SARS-CoV-2. To accession samples, racks are scanned on a Zianth flatbed scanner. The scanner output file is uploaded to the LIMS and barcodes become linked to well positions on the plate. Barcodes are matched against expected barcode IDs from the eTRFs; any unmatched sample tubes are assigned "missing information" status and manually removed from the rack, and the entire rack is rescanned until there is a perfect match to the eTRF database. For pooled samples, a new parent rack containing clean tubes is scanned and every child rack is associated with the parent rack via a barcode scanning step prior to removing an aliquot of the sample. A rack ID from each child plate is also associated with the parent plate. The LIMS generates a plate definition file containing the barcode

IDs for each sample which is loaded into the Design and Analysis software that drives the QuantStudio 6 Pro qPCR (ThermoFisher). Following qPCR the results file is ingested into LIMS and the logic table is applied to each sample, resulting in a positive, negative, inconclusive, or invalid call. Following review by licensed clinical laboratory scientists, results are reported to medical providers and state and local health agencies via secure HL7 messaging.

## Results

### A campus-wide testing plan

Recent modeling suggests that testing frequency is critical for reducing community transmission; testing at least once per week is needed to stop sustained transmission of SARS-CoV-2 under pre-pandemic conditions [3]. At full capacity, UC Santa Cruz has nearly 21,000 students, staff, and faculty. To help mitigate the risk of SARS-CoV-2 transmission we aimed to establish a twice weekly testing program for our entire campus population, which would require 7,000 samples being tested each day. We established a network of collection sites across campus at easily accessible, large, well-ventilated spaces. An electronic test requisition form is generated when participants check in at a testing site and link their protected health information (PHI) to a sample collection tube via a 1D barcode scanner. Participants then perform a supervised self-collection of their nasal specimen, deposit the sample tube in a rack and dispose of the swab in a biohazard waste container. Specimen racks are returned to the laboratory at the end of the day for processing.

### Automated, high throughput COVID-19 testing

Using an Agilent Bravo NGS-A liquid handler, we developed an automated, parallel RNA extraction protocol for nasal swab specimens. 1.4 mL sample collection tubes can be racked into the standard Society for Biomolecular Screening (SBS) 96-well format for laboratory automation. This enables parallel processing of 93 samples and 3 controls per plate. Control samples consist of a positive extraction control containing HEK 293 cell lysate in transport media spiked with synthetic SARS-CoV-2 RNA (Twist Biosciences) at 10x the analytical limit of detection (LOD), a negative extraction control consisting of only HEK 293 cell lysate in transport media, and a no-template control consisting of ddH2O. These three samples are carried through the entire extraction process.

We implemented a multiplex RT-qPCR assay developed at the Center for Infection and Immunity, Columbia University (EUA#200510). This high sensitivity assay (650 copies/mL) amplifies two regions of the nucleocapsid (N) gene as well as the human mitochondrial RNase P (RP) transcript (Table 1). We opted for a high sensitivity assay in order to facilitate pooled sample testing (see below). Using SARS-CoV-2-negative or -positive specimens obtained from symptomatic patients or asymptomatic carriers, we determined the positive and negative percent agreement for the multiplex assay to be greater than 97% and 95%, respectively (Tables 2 and 3, respectively). The results of our LDT were validated against a high sensitivity SARS-CoV-2 test from Pangea Laboratories.

At low prevalence, sample pooling can greatly accelerate high capacity testing for SARS-CoV-2 [24–27]. We took advantage of the high-sensitivity multiplex assay described above and the 96 channel pipetting capacity off the Bravo NGS-A platform to develop a sample pooling protocol (Fig 2A). In this scheme, individual sample racks (child racks) are combined into a new rack of 96 matrix tubes (parent rack). Each barcoded child rack is associated with a single barcoded parent rack and each child sample tube with its unique rack position (ie. position C4) is associated with a new barcoded parent tube with the identical rack position in the parent rack. Construction of sample pools are managed by the LIMS, which enables accessioning of

**Table 1. The limit of detection for the UC Santa Cruz Multiplex SARS-CoV-2 assay.**

| Dilution (vg/uL) | N1 positive | N2 positive | Cq Mean N1 | Cq Mean N2 | Cq Mean RNase P |
|---|---|---|---|---|---|
| 10 | 3/3 | 3/3 | 31.66 | 30.89 | 25.02 |
| 5 | 3/3 | 3/3 | 32.69 | 31.84 | 25.10 |
| 2.5 | 3/3 | 3/3 | 33.98 | 33.33 | 25.11 |
| 1.25 | 3/3 | 3/3 | 34.25 | 33.50 | 24.91 |
| **0.625** | **3/3** | **3/3** | **35.18** | **34.63** | **24.99** |
| 0.3125 | 2/3 | 2/3 | 36.46 | 35.41 | 24.99 |
| 0.156 | 1/3 | 1/3 | 37.33 | 36.89 | 24.65 |
| 0.078 | 0/3 | 0/3 | ND | ND | 24.39 |
| 0.039 | 1/3 | 1/3 | 37.18 | 35.90 | 24.93 |
| 0 | 0/3 | 0/3 | ND | ND | 24.65 |

the parent rack and parent tubes, as well as the association of each child rack and child tube. This process ensures traceability of individual samples. We refer to this strategy as "linear pooling" because the position in the 96 well array determines which samples from each child rack contribute to the pooled sample. This approach is in contrast to reported "matrix" pooling strategies which employ a more complex pool building algorithm in order to deconvolute redundant pools through two PCR reactions [27]. In our linear pooling approach, sample deconvolution occurs by re-testing the individual child samples that make up a positive pool. Our laboratory validated 10:1 sample pooling following FDA guidelines (Fig 2B–2D). The Ct values for N1 and N2 in the parent (pooled) or children (individual) samples are highly correlated (R = 0.96 and 0.95, respectively) with an offset of ~ 3.3 Ct in the pooled samples, as expected for a 10-fold dilution of a positive sample (Fig 2B and 2C). The sensitivity and specificity of the pooled assay was also validated by the Pangea Labs test (Fig 2D and Table 4). The overall performance of the test on both unpooled and pooled samples has been robust. Fig 2E compares cycle thresholds for N1, N2 and RP, from >10,000 pooled samples, >10,000 unpooled surveillance samples and >30,000 clinical samples. We found no significant difference in the distribution of N1 or N2 cycle thresholds from positive clinical or individual surveillance samples. By contrast we observed significant differences in the cycle thresholds for N1 and N2 from pooled sample tests as compared to unpooled surveillance and clinical samples. This result was expected as positive samples are diluted approximately 10-fold relative to unpooled tests. We observed a slight, but significant difference in Ct values for RP from negative unpooled surveillance samples and negative clinical samples (mean Ct 24.77 and 25.10, respectively). This difference could be due sample collection by a healthcare provider as compared to supervised-self collection of surveillance samples. Finally, we calculated the

**Table 2. Validation of UC Santa Cruz Multiplex SARS-CoV-2 assay by a commercial lab using true positive and true negative clinical samples.**

| Samples Tested Individually | Comparator Method Result | |
|---|---|---|
| Candidate Test Result | Positive | Negative |
| Positive | 30 | 0 |
| Negative | 0 | 30 |

(samples collected on symptomatic patients).

Performance of the UCSC Multiplex SARS-CoV-2 Assay against the Pangea comparator: Positive Percent Agreement: 30/30 = 100% (95%CI: 90%-100%), Negative Percent Agreement: 30/30 = 100% (95%CI: 90%-100%).

**Table 3. Validation of UC Santa Cruz Multiplex SARS-CoV-2 assay by a commercial lab using true positive and true negative asymptomatic surveillance samples.**

| Samples Tested Individually | Comparator Method Result | |
|---|---|---|
| *Candidate Test Result* | *Positive* | *Negative* |
| **Positive** | 29 | *0* |
| **Negative** | 1 | *30* |

(samples collected on asymptomatic patients).

Performance of the UCSC Multiplex SARS-CoV-2 Assay against the Pangea Comparator: Positive Percent Agreement: 29/30 = 97% (95%CI: 84%-100%), Negative Percent Agreement: 30/30 = 100% (95%CI: 90%-100%).

turnaround time for all COVID-19 tests performed during the fall and winter quarters. Fig 2F shows the distribution of turnaround times (defined as the total time between sample accessioning and reporting) with a mean of ~25 hours and a standard deviation of ~11 hours. A significant fraction of the test results were returned in less than 12 hours.

## Laboratory information management system (LIMS)

The blueprint developed by our colleagues at the Integrated Genomics Institute (IGI) at UC Berkeley [22] clearly demonstrated that a LIMS is required for sample accessioning, tracing and reporting. Because the liquid handling strategy and sample collection tubes differed from the IGI blueprint, it was necessary to develop a custom LIMS, rather than licensing an established LIMS. Like the IGI, our LIMS consists of three modules (Fig 3). The accessioning module matches sample tube barcodes to electronic test requisitions that are deposited into the LIMS from the sample collection kiosks. The test requisitions are received electronically by the LIMS. When sample racks arrive in the lab they are heat inactivated (30 minutes at 70ºC) [28] then the 2D barcodes for all 93 samples are scanned en masse using a flatbed scanner (Ziath). The scanner output file is uploaded to the LIMS and barcodes on the tubes are matched by the LIMS against the barcode associated with the electronic test requisition form for each participant. If a sample barcode cannot be matched against a complete requisition, that sample is flagged by the LIMS and removed from the rack. The accessioning process is then repeated for the entire rack. For pooled sample testing, child racks are scanned and associated with a rack of clean 1.4 mL barcoded tubes in a barcoded parent rack. A fraction of each sample from a child rack is transferred to the parent rack in order to build the pools for testing. Each tube within the parent rack is associated with up to ten individual child samples. The parent plate is scanned and uploaded to the LIMS. Each sample extraction plate, RNA storage plate, and qPCR plate is barcoded and associated with each sample throughout the process, ensuring traceability. The final module links qPCR results to individual samples and applies a logic table to call results as detected, not detected, inconclusive, or invalid. The LIMS enables rapid and facile review by clinical laboratory scientists and timely result reporting to healthcare providers and local and state officials via HL7.

## LIMS integration with public health and provider portals allowed scaling

The original implementation of the LIMS had an entirely manual intake of orders and outflow of results. For the kiosks, a flat file was created that could be imported into the LIMS to create new patient and order entries. For reporting results, a pdf result for each patient was generated to send to the provider using a flat file and mail merge, and a flat file was generated for mandatory reporting to the State of California. This minimum-viable-product met requirements but

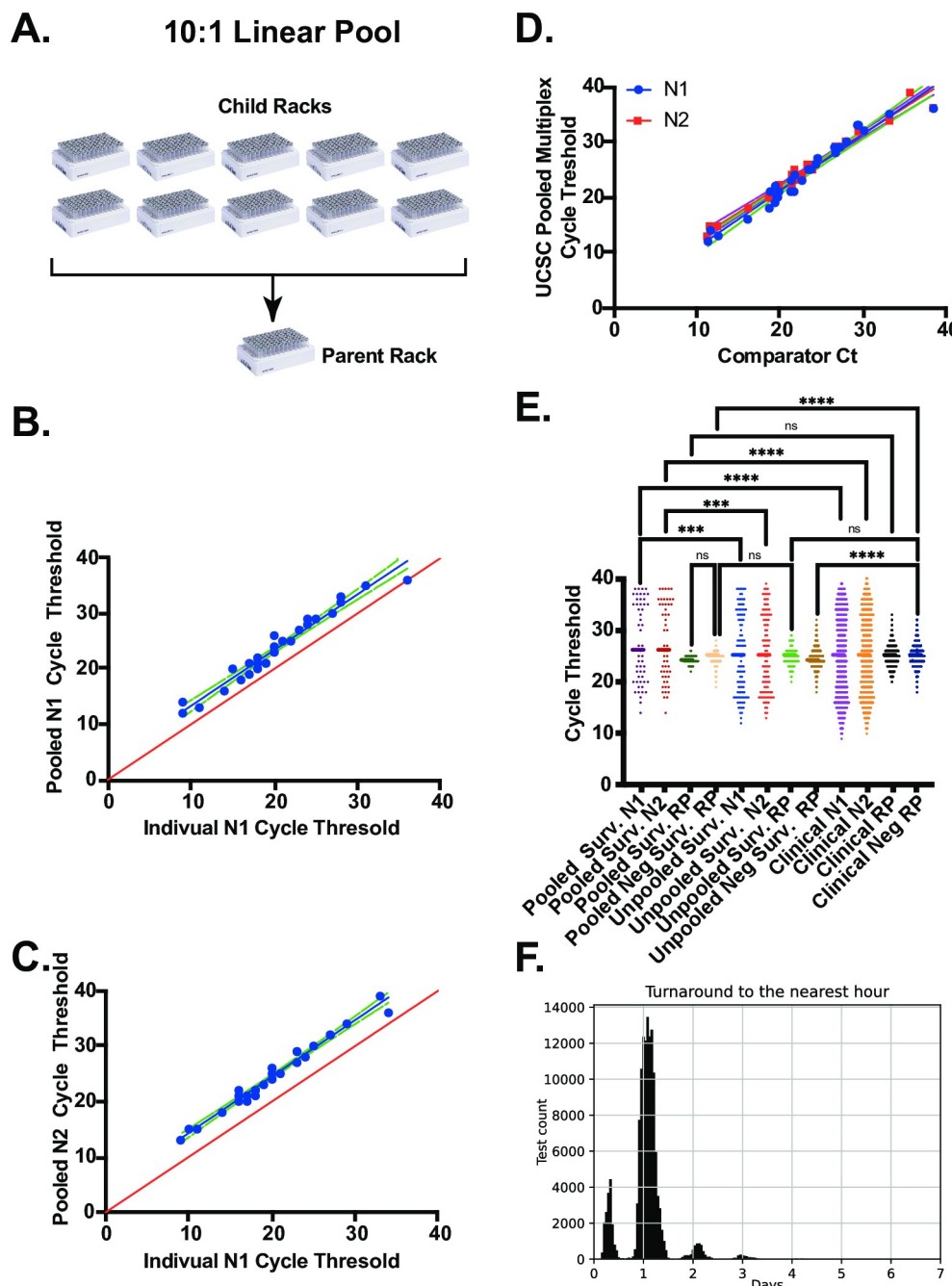

**Fig 2. Validation of a pooled sample testing strategy for campus-wide SARS-CoV-2 surveillance.** (A) 10:1 linear pooling strategy. Upto 10 individual sample racks (children) are combined into a single pooled sample (parent) rack. (B) UCSC multiplex N1 assay on 30 individual or pooled positive samples. (C) UCSC multiplex N2 assay on 30 individual or pooled positive samples. (D) Analysis of pooled positive samples with the UCSC multiplex or comparator assay. N1 amplicon (blue) and N2 amplicon (red). (E) Overall performance of the UCSC multiplex assay on thousands of clinical, surveillance and pooled surveillance samples. (F) Turnaround time for both clinical and surveillance samples, rounded to the nearest hour.

**Table 4. Validation of UC Santa Cruz Multiplex SARS-CoV-2 assay by a commercial lab using true positive and true negative pooled surveillance samples.**

| Samples Tested 10-Sample Pool | Comparator Method Result | |
|---|---|---|
| Candidate Test Result | Positive | Negative |
| Positive | 29 | 1 |
| Negative | 1 | 29 |

(samples collected on symptomatic patients).

Performance of the UCSC Multiplex SARS-CoV-2 Assay on 10-sample pools against the Pangea comparator: Positive Percent Agreement: 29/30 = 97% (95%CI: 84%-100%), Negative Percent Agreement: 29/30 = 97% (95%CI: 84%-100%).

required substantial daily human intervention. In order to scale up, we implemented an automatic report delivery system integration by contracting with middleware Software-As-A-Service (SAAS) providers (BridgeConnect and Santa Cruz Health Information Exchange). This integration connected the LIMS with the existing systems at the County and State, and with the patient medical portal for approval and communication of results and accessioning of samples from the kiosk sites into the LIMS (Fig 4). It also enabled Accessioning and Resulting to be performed within the existing electronic medical record (EMR) software Point and Click (PNC). Another piece of the integration was creating a staff demographic feed to allow them to be tested within PNC. This iterative approach to integration, with multiple plans, allowed the project to be launched quickly and then later improved so that it could scale. By careful collaboration across various parts of campus we brought together technical experts from genomics, student health, and ITS to provide a seamless experience to patients and students in the community. This information management solution enabled automated test requisitioning, sample accessioning and result reporting and was critical to scaling-up testing capacity.

## Potential impact of surveillance on campus-wide SARS-CoV-2 transmission

UC Santa Cruz opened the 2020 academic year under fully remote instruction with low on-campus student density. Approximately 1200 students lived on campus during Fall quarter; students were tested upon arrival and initially on a twice weekly cadence thereafter. Fig 5A shows the number of daily tests performed during the academic year. The methods described above enabled an efficient scale up of COVID-19 surveillance during the late fall and early winter quarters. During the Fall 2020 quarter we observed an increase in new cases per day during November and December (Fig 5B). Entry testing upon the return of students to campus in the Winter 2021 also discovered a large number of COVID-19 positive individuals. The rapid isolation of positive students coincides with the rapid decline in daily cases and the campus positivity rate (Fig 5C). Between the peak on January 4th and February 1st, the positivity rate declined ~17 fold, to 0.09%. Although there were 227 individuals who tested positive for SARS-CoV-2 between July 14, 2020 and May 1, 2021, case investigations found no evidence of transmission on the UCSC campus. We also compared the positivity rate for samples collected on campus to those that were collected off campus by safety net health care providers and processed by the UC Santa Cruz Colligan Clinical Diagnostics Laboratory (CCDL). Fig 5D shows that compared to symptomatic clinical samples, the positivity rate for samples collected through the asymptomatic testing plan is significantly lower than samples collected from the broader community, as expected. Additionally, recent work from the University of California

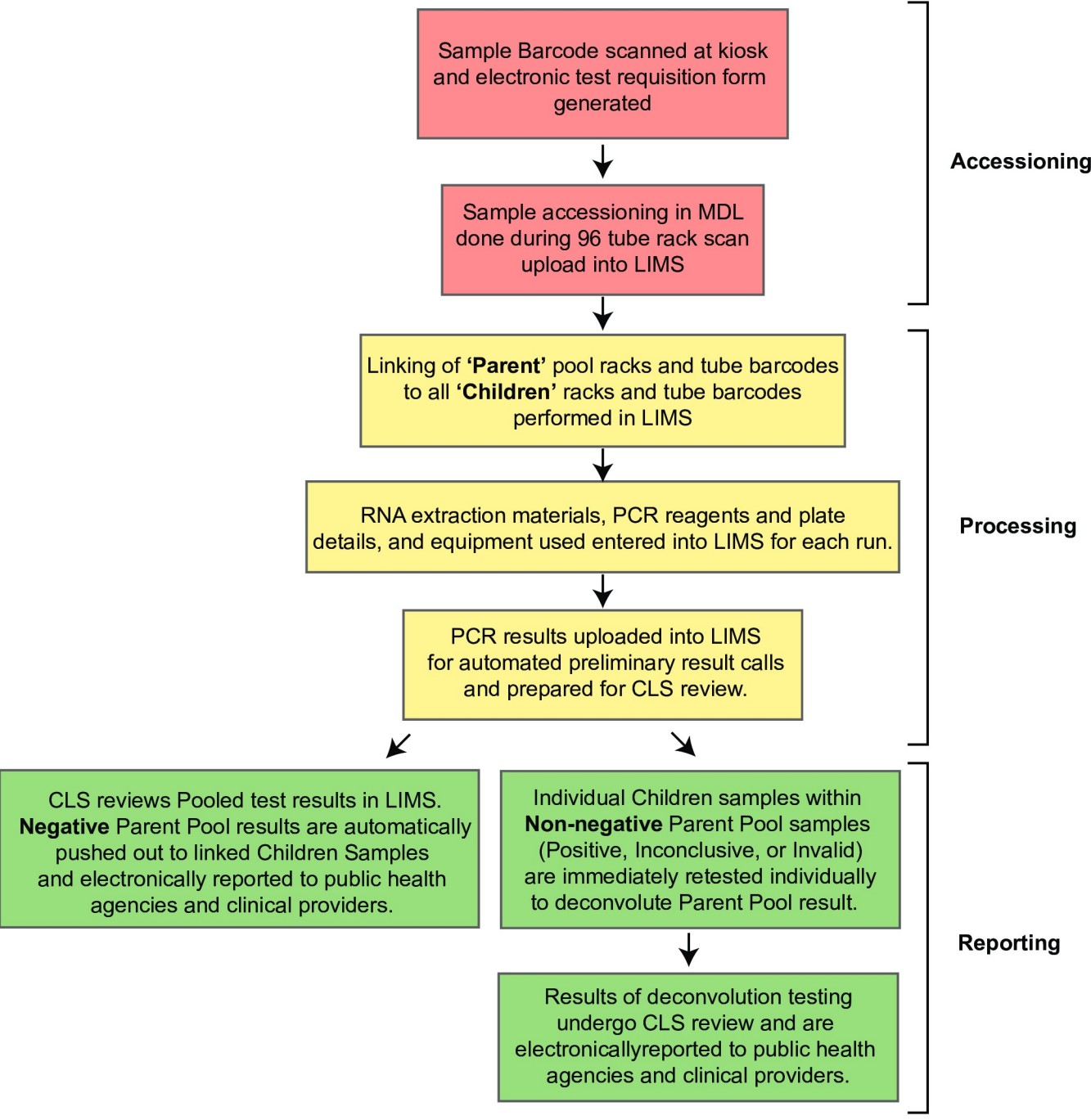

**Fig 3. Overview of sample management by the custom laboratory information management system.** The workflow is divided into three basic modules: sample accessioning, sample processing and result reporting. To accelerate testing, accessioning and reporting steps are fully automated.

COVID-19 Task Force compared the COVID-19 incidence of 20–29 year olds on campus and the surrounding community. The COVID-19 incidence was approximately 50% lower on the UCSC campus population than in the surrounding county 20–29 year-olds, suggesting that the multi-layered mitigation measures, including surveillance testing, limited SARS-CoV-2 transmission on campus [29].

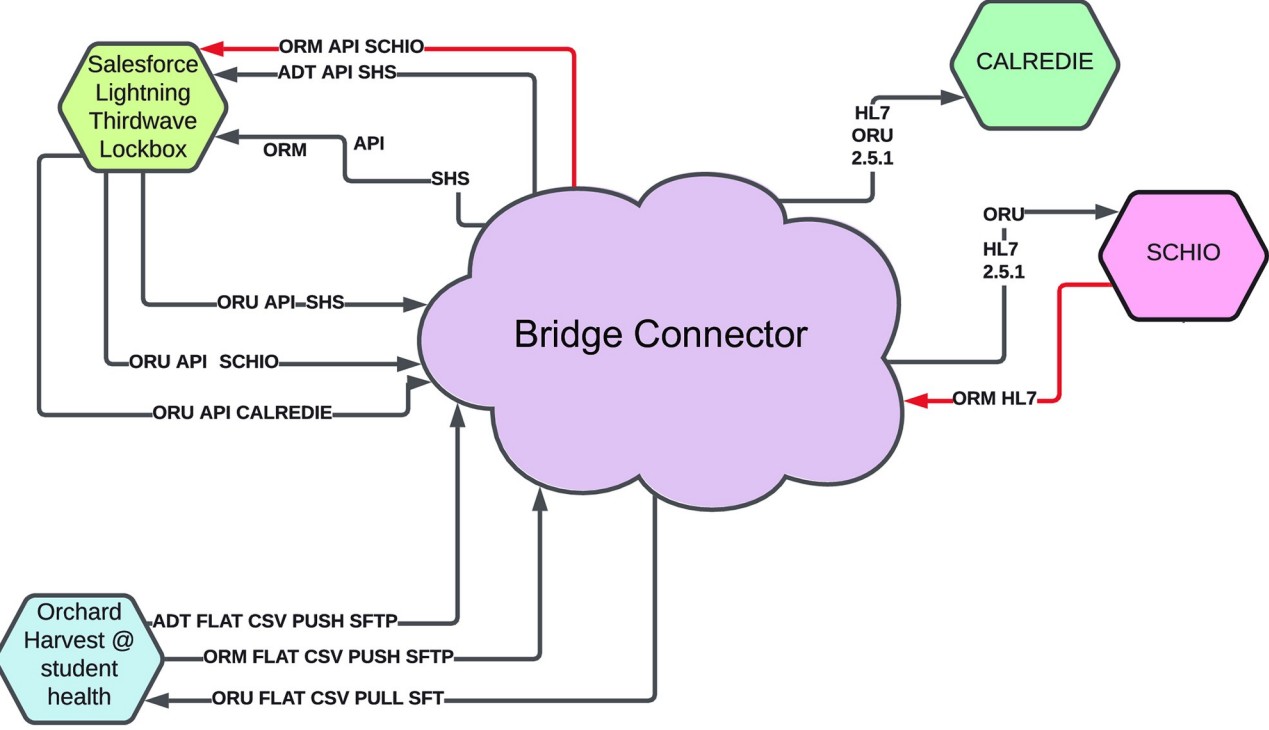

**Fig 4. Integration overview between salesforce thirdwave LIMS at UCSC CCDL, Orchard Harvest at student health, Calredie (California department of public health), and SCHIO for third party integrations in the community.** (A)Orchard Harvest, the Laboratory Information System in student health services. Used for orders originating from UCSC affiliates and for communicating results to affiliates. (B) Salesforce Lightning Thirdwave Laboratory Information System. Used by clinical laboratory for processing samples and results. '(C)' Integration platform which converts orders from flat file from Harvest to API Calls in salesforce, and converts results from API in salesforce to flat file for Harvest ingestion. Also exports data to the CDPH in HL7 and to SCHIO for community partner results. (D) Calredie is the CDPH platform for communicating results and orders. (E) Santa Cruz Health Information Exchange—non-profit integration platform provider for community to share data. (F) to be built—order interface from third party community providers directly into the Thirdwave LIMS without having to user provider portal.

## Discussion

Even after a full year of the COVID-19 pandemic, scaling up testing remains a major challenge for many institutions [17]. Pooled testing is recognized as an important approach for SARS-CoV-2 surveillance [30–33]. We believe our implementation represents a robust process that is applicable on an institutional level. The approach described here leverages barcoded, SLAS-compatible sample collection tubes to facilitate hands-free accessioning and reporting. To increase testing capacity and conserve resources, we implemented a sample pooling strategy. This approach also accelerates turnaround times and increases throughput as prevalence drops. Finally, a custom LIMS enables sample traceability throughout the testing process as well as deconvolution of pooled samples.

The COVID-19 testing and surveillance efforts at UC Santa Cruz were inspired by the work of our colleagues at other University of California Campuses. However, the relative geographic isolation of Santa Cruz county, combined with limited high complexity diagnostic infrastructure and expertise, drove us to seek solutions that could allow our testing program to scale and remain resilient to supply chain disruptions. The result was a strategy that differed significantly from our colleagues at the IGI Testing Consortium, which provided the blueprint that many labs, including our own, initially followed [22]. One of the key initial steps was to create a laboratory developed test using reagents that had not already received Emergency Use

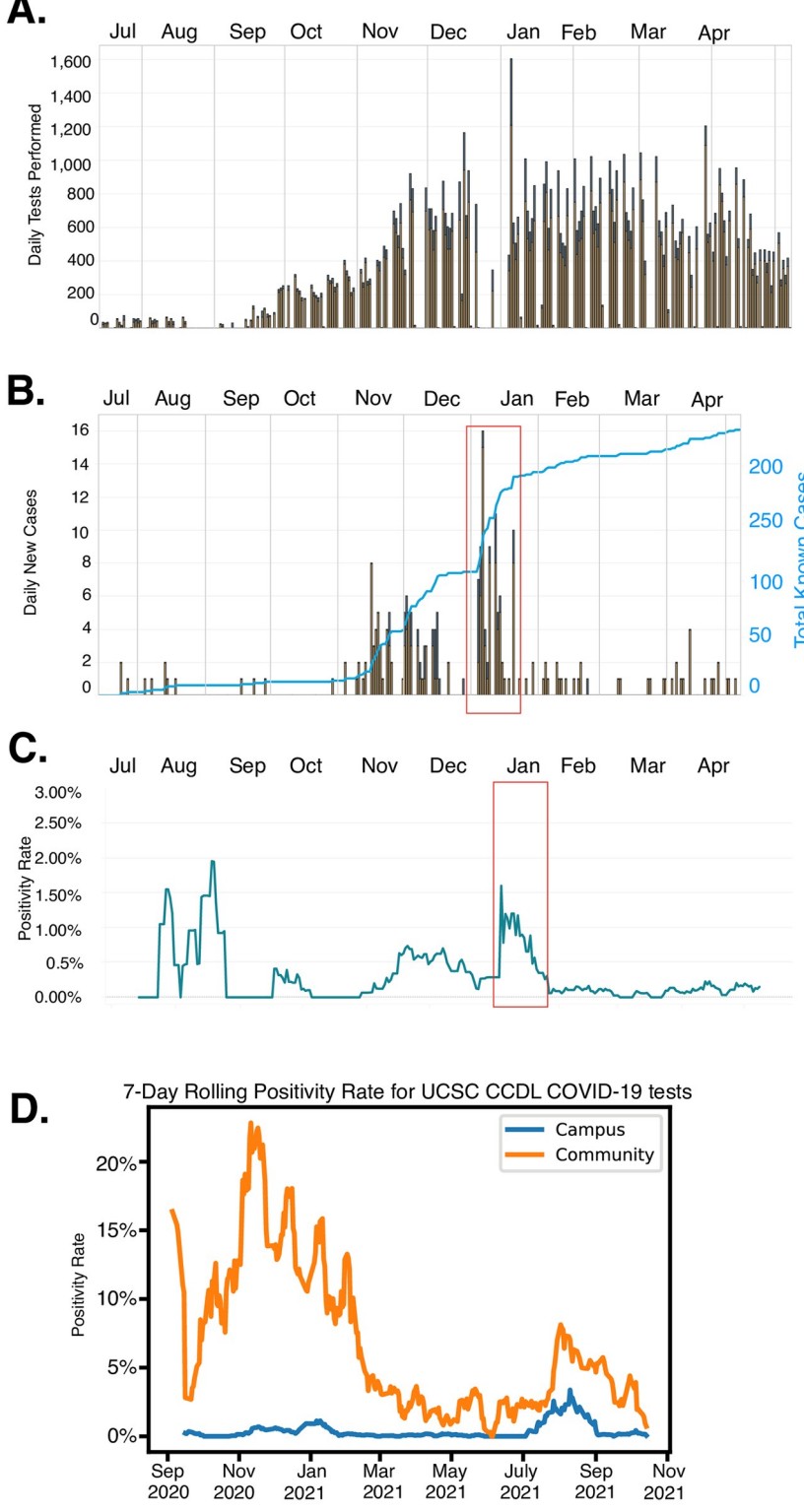

**Fig 5. Scaling and impact of asymptomatic testing at UC Santa Cruz during fall 2020 and winter 2021 academic quarters.** (A) Increase in tests performed per day at the UC Santa Cruz Molecular Diagnostic Lab. Tests for students represented by gold bars, tests for staff in blue. (B) New cases reported on campus per day for students (gold bars) and staff (blue bars) and cumulative case count (blue line). (C) 7 day rolling average positivity rate for UC Santa Cruz students and staff. Red box signifies a potential spike in positive cases due to imported infections as students

repopulated the campus after the winter break. (D) Comparison of positivity rate for clinical samples collected by local health care providers and symptomatic and asymptomatic samples from the UC Santa Cruz campus. All data are available at our campus COVID-19 dashboard https://recovery.ucsc.edu/reporting-covid/covid-tracking/ and are updated in real-time.

Authorization (EUA) from the FDA. This approach allowed us to avoid potential bottlenecks without depleting reagents that were needed by molecular diagnostic labs at medical centers. Instead, we developed an RNA extraction system using research grade reagents from commercial labs. We used a robust one-step RT-PCR reaction mix that was closely related to the gold standard Taq-Path reagent, but was not a component of any test with EUA. We also implemented a multiplex test developed at the Center for Immunity and Infection at Columbia University. This multiplex assay enabled the sensitive detection of SARS-CoV-2 and the endogenous RP transcript, which is an important indicator of sample quality and allowed for pooled sample testing.

Sample collection, accessioning, and result reporting are among the major challenges in scaling up a surveillance program. One important innovation in our process was the use of Society of Laboratory Automation and Screening (SLAS) compatible 1.4 mL barcoded sample collection tubes with 0.6 mL transport medium rather than larger collection tubes, such as 15 mL falcon tubes (Fig 1). The 1.4 mL sample tubes are critical because they enable hands-free sample accessioning, processing and reporting. Each 1.4 mL tube is labeled with machine-readable 1D and 2D barcodes (on the side and bottom, respectively) that encode an identical numerical identifier. The sample collection tubes are racked into 96 tube arrays, compatible with an automated 96 screw-cap capper/decapper and the Agilent Bravo NGS-A liquid handler 96-well pipetting head. This system eliminated the need for both manual evaluation and recording of sample identifiers and manual transfer of the sample into 96-well extraction plates. The NGS-A's 96 channel pipette head also enables a linear pooling strategy. In this case pooled (parent) sample racks are assembled from up to 10 individual sample racks (children). This process requires a single box of tips per child rack to prevent cross contamination of the individual samples. We found that when assembling parent racks it was important to keep the pipetting head in a fixed position relative to the sample racks in order to avoid carry over contamination during pooling (data not shown). Using this method, a single NGS-A is capable of pooling and extracting 930 samples in approximately 1 hour. This process scales efficiently with additional liquid handlers and staffing. In the event of a positive pool deconvolution occurs by simply retesting individual samples that contributed to the pool.

Like our colleagues at UC Berkeley's IGI, we also developed a custom laboratory information management system (LIMS) using the SalesForce platform. While our initial decisions were informed by the blueprint paper, the key differences described above (1.4 mL matrix tubes, sample pooling and deconvolution) required a customized LIMS that was distinct from the IGI platform. Although development of a custom LIMS was laborious, a fully integrated system for accessioning, tracking and deconvoluting sample pools, as well as reporting results was the single most important element in scaling testing capacity. This system completely eliminated the need for paper test requisition forms and any human readable identifiers.

## Conclusions

We describe a process for hands-free accessioning, sample pooling, automated extraction, pool deconvolution and reporting that can be completed in a single day. In theory, a modestly equipped and staffed lab can efficiently process thousands of pooled samples per day. This

system enables efficient surveillance for SARS-CoV-2 or pathogens linked to future pandemics on an institutional scale.

## Supporting information

**S1 Data.**
(XLSX)

## Acknowledgments

We would like to thank JoeBen Bevirt (Joby Aviation) and Dr. Nischay Mishra (Columbia University) for technical expertise in automation and SARS-CoV-2 molecular diagnostics, respectively. We thank our colleagues at the UCSF/CZI Biohub Drs. Joe DeRisi and Stephen Miller for testing our clinical and surveillance samples while our lab was briefly shut down during the CZU Lightning Complex Fire evacuations. We thank the staff of the Colligan Clinical Diagnostic Lab (formerly known as the UCSC Molecular Diagnostics Lab) for critical comments on the manuscript. We thank Dr. Susan Weaver and Robert Gosselin for their support on clinical laboratory practice and compliance.

We thank our Santa Cruz County community partners, including Salud Para La Gente, Santa Cruz Community Health, The Santa Cruz County Health Department, Encompass, the Santa Cruz County Jail, Santa Cruz County Probation, and the Santa Cruz County Community Foundation for ongoing collaboration and support throughout the pandemic. Special thanks belong to Santa Cruz Community Health and Salud Para La Gente, both federally qualified health centers, for their partnership in developing and improving our testing lab assays and protocols. Santa Cruz Community Health was our first partner, and assisted in several validation efforts and provided valuable feedback as we designed our testing service protocol and reporting structure. We appreciate the individual efforts of Santa Cruz Community Health CEO Leslie Conner, Medical Director Dr. Casey Kirkhart, COO Philippa Barron, and Operations Director, Stephanie Macwhorter. Salud Para La Gente similarly partnered with us on efforts to continuously improve our COVID-19 testing and was our lead partner in thinking through adoption of AN swabs. This collaboration resulted in a significant positive impact for our community and our campus by improving the swabbing experience and enabling rapid ramp up of testing due to the increased ease of AN swab use. We appreciate the individual efforts of Salud Para La Gente CEO Dori Rose Inda, Medical Director Dr. Amy McEntee, Director of Nursing Danielle Obinger, Jennifer Wood, Senior Systems Analyst, and Associate Director of Nursing Iriana Hinman.

## Author Contributions

**Conceptualization:** Auston M. Kilpatrick, Michael D. Stone, Jeremy R. Sanford.

**Data curation:** Jeremy R. Sanford.

**Investigation:** Terren Chang, Jolene M. Draper, Anouk Van den Bout, Ellen Kephart, Hannah Maul-Newby, Yvonne Vasquez, Savanna Randi, Martina Pedersen, Maeve Nave, Scott La, Natalie Gallagher, Molly M. McCabe, Namrita Dhillon, Frank Dang.

**Methodology:** Terren Chang, Jolene M. Draper, Anouk Van den Bout, Ellen Kephart, Hannah Maul-Newby, Yvonne Vasquez, Jason Woodbury, Savanna Randi, Namrita Dhillon, Isabel Bjork, Michael Luttrell, Frank Dang, Elizabeth Miller, Olena Vaske, Michael D. Stone, Jeremy R. Sanford.

**Project administration:** Jeremy R. Sanford.

**Supervision:** John B. MacMillan, Ralph Green, Elizabeth Miller, Olena Vaske, Michael D. Stone, Jeremy R. Sanford.

**Visualization:** Michael D. Stone, Jeremy R. Sanford.

**Writing – original draft:** Auston M. Kilpatrick, Michael D. Stone, Jeremy R. Sanford.

**Writing – review & editing:** Terren Chang, Jolene M. Draper, Anouk Van den Bout, Ellen Kephart, Hannah Maul-Newby, Isabel Bjork, Auston M. Kilpatrick, Olena Vaske, Michael D. Stone.

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
