## [Decision Letter · Decision Letter 0]

2 Sep 2021

PONE-D-21-23418

A method for campus-wide SARS-CoV-2 surveillance at a large public university

PLOS ONE

Dear Dr. Sanford,

Thank you for submitting your manuscript to PLOS ONE. After careful consideration, we feel that it has merit but does not fully meet PLOS ONE’s publication criteria as it currently stands. Therefore, we invite you to submit a revised version of the manuscript that addresses the points raised during the review process.

In particular, the reviewers request clarification of some experimental procedures and statistical analysis.

We look forward to receiving your revised manuscript.

Kind regards,

David M. Ojcius

Academic Editor

PLOS ONE

Journal Requirements:

“R35GM130361

GM095850

The work was funded by the Office of Research at UC Santa cruz”

“I have read the journal's policy and the authors of this manuscript have the following competing interests: Jeremy Sanford and Michael Stone are paid consultants and have an ownership interest in SummerBio, a commercial COVID testing laboratory that specializes in institutional surveillance.”

We note that one or more of the authors are employed by a commercial company: name of commercial company.

Reviewers' comments:

Reviewer's Responses to Questions

**Comments to the Author**

1. Is the manuscript technically sound, and do the data support the conclusions?

Reviewer #1: Yes

Reviewer #2: Yes

2. Has the statistical analysis been performed appropriately and rigorously? 

Reviewer #1: N/A

Reviewer #2: Yes

3. Have the authors made all data underlying the findings in their manuscript fully available?

Reviewer #1: No

Reviewer #2: Yes

4. Is the manuscript presented in an intelligible fashion and written in standard English?

Reviewer #1: Yes

Reviewer #2: Yes

5. Review Comments to the Author

Reviewer #1: -There are several references that are not pre-prints or peer-reviewed publications (e.g., 27) and Ref. 12 has no information besides a link. Also, please check formatting for things like FDA documents and laws (CLIA). It is unlikely that they are best cited as a URL.

-Many aspects of your introduction remind me of a paper out of UC Berkeley from June 2020 (https://www.nature.com/articles/s41587-020-0583-3), also covered in Francis Collin's Blog https://directorsblog.nih.gov/2020/05/12/pop-up-testing-lab-shows-volunteer-spirit-against-deadly-pandemic/. Nevertheless, this paper is scarcely cited. It looks like others have been published as well... https://www.jbc.org/article/S0021-9258(17)50379-8/fulltext & https://www.nature.com/articles/s41564-020-00818-3 & https://journals.plos.org/plosone/article?id=10.1371/journal.pone.0251296. This manuscript would benefit from a more thorough literature review and diligent referencing.

-Instead of referring to "Many studies" in "Many studies have suggested that oral swabs represented a facile self-collection alternative to NP swabs compatible with mass testing [24]", it may help to clarify that it was a meta-analysis (e.g., "A meta-analysis of __ studies suggests that..."

-We have practically no information regarding how the comparison studies were carried out. Were these tests carried out under an IRB? If not, what medical justification was there to collect both samples from patients? If so, what is the protocol #? What is the time difference between OP/NP samples? What are the characteristics of this population?

-What sort of statistical analysis did you do to arrive at the statement "After careful consideration of the literature and further validation studies in our lab, we implemented a supervised self-collected anterior nares (AN) nasal

specimen." Are the "further validation studies" described anywhere?

-Who were the "healthy donors" in "We performed a comparator study using healthy donors to determine if sample integrity was maintained in the 1.4 mL tubes compared to the 15 mL sample collection tubes." What sorts of protocols were used for recruitment, informed consent, etc?

-Since there are already "blueprint" papers out there like this, it would be helpful to see how the methods outlined here compare to the previously published efforts. Currently, the manuscript reads almost like a methodology paper without comparison to existing methods.

-Please make sure that the claim "Although there are many reports of the theoretical implementation of sample pooling approaches for COVID-19 surveillance testing [33]; [34]; [35], there are few reports showing how this

process has been put into practice on an institutional level." is accurate. There are reports I'm aware of that describe this.

-Is the LIMS you built truly the result of only a collaboration between your group and Third Wave Analytics? ("A custom laboratory information management system (LIMS) was developed in collaboration with Third Wave Analytics (San Francisco) using the Salesforce Lightning Platform and Experience Cloud.") Reading over the June 2020 paper from UC Berkeley, it looks like they also built such a system with Third Wave Analytics. If the system you built was at all an adaptation of this or a system another group built, it would be prudent to cite that group.

-The section entitled "Impact of surveillance on campus-wide SARS-CoV2 transmission" is somewhat misleading. No epidemiological methods were employed in this paper to demonstrate that surveillance done affected overall transmission. We are not provided with any of the surveillance parameters for non-student groups (e.g., frequency, % of population of interest), nor prevalence in the local community.

-Turn around time is mentioned in the introduction, but we are provided with no actual data about it in the body of the manuscript. Please provide analysis with the raw data.

-Please check figure #s before resubmission.

As a final note, the document has inconsistent formatting with paragraph breaks an indentation. It would be appreciated if revisions were properly formatted for ease of navigation.

Reviewer #2: Overall this is a well written and clearly presented paper that does not require major revisions. It describes the practical implementation of a pooled testing strategy for automated handling, testing and tracking of a high volume of self-collected COVID-19 nasal swab samples, and should be of interest to audiences planning to implement or optimize strategies for high volume viral testing. While pooled strategies for campus COVID-19 surveillance have been widely mentioned in nontechnical news reports, this manuscript provides both technical details of implementation and useful supporting research that was conducted in order to validate the strategy and to optimize protocols.

In the results section, the statement "Surprisingly we observed a slight, but significant difference in Ct values for RP from negative individual surveillance samples and negative clinical samples (mean Ct 24.77 and 25.10, respectively)." remains as a bit of a loose end that needs to be explained more completely. The expectation (and observation) in the pooled surveillance samples was that there would be a difference in the two Ct values due to dilution -- why is there a different expectation for RP and what might be an explanation for the observed results?

The abbreviation RP does not seem to be defined at its first occurrence.

Figure 1 might be less cluttered if presented as box and whisker plots, and I don't think the explicit connection between sample pairs is necessary to the point of the figure.

Figure 2 labels contain misspellings of the word "Individual".

Are automation programs for the robotics and custom LIMS code available anywhere e.g. on a GitHub? It seems like those might be useful for other teams trying to implement similar protocols or using the same hardware for related purposes.

6. PLOS authors have the option to publish the peer review history of their article (what does this mean?). If published, this will include your full peer review and any attached files.

Reviewer #1: No

Reviewer #2: No

---

## [Author Response · Author response to Decision Letter 0]

23 Nov 2021

Specific points

1. Reviewer #1: -There are several references that are not pre-prints or peer-reviewed publications (e.g., 27) and Ref. 12 has no information besides a link. Also, please check formatting for things like FDA documents and laws (CLIA). It is unlikely that they are best cited as a URL.

All citations have been updated and are now in the correct format.

2. Many aspects of your introduction remind me of a paper out of UC Berkeley from June 2020 (https://www.nature.com/articles/s41587-020-0583-3), also covered in Francis Collin's Blog https://directorsblog.nih.gov/2020/05/12/pop-up-testing-lab-shows-volunteer-spirit-against-deadly-pandemic/. Nevertheless, this paper is scarcely cited. It looks like others have been published as well... https://www.jbc.org/article/S0021-9258(17)50379-8/fulltext & https://www.nature.com/articles/s41564-020-00818-3 & https://journals.plos.org/plosone/article?id=10.1371/journal.pone.0251296. This manuscript would benefit from a more thorough literature review and diligent referencing.

We indeed drew inspiration from our colleagues at UC Berkeley. We now include a broader array of references for other campus testing programs as well.

3. Instead of referring to "Many studies" in "Many studies have suggested that oral swabs represented a facile self-collection alternative to NP swabs compatible with mass testing [24]", it may help to clarify that it was a meta-analysis (e.g., "A meta-analysis of __ studies suggests that...

4. We have practically no information regarding how the comparison studies were carried out. Were these tests carried out under an IRB? If not, what medical justification was there to collect both samples from patients? If so, what is the protocol #? What is the time difference between OP/NP samples? What are the characteristics of this population?

All of the work was conducted by our diagnostic lab, operating under the license of the UCSC Student Health Center, which is subject to the regulatory framework that governs diagnostic laboratories, including CLIA guidelines. The comparison studies do not qualify as university research. IRB review was not necessary for this work as it focused on validating and improving the quality of our diagnostic test and was performed according to CLIA guidelines. The data was not generated for and cannot be used for university or other research; summary data is presented in this paper with the sole purpose of describing the process of validation and test improvement that we underwent in order to maximize text reliability, accuracy and value. Sample collection was performed by our clinical partners from health care providers (UCSC Student Health Center, Salud Para La Gente and Santa Cruz Community Health Centers) after review by their respective ethics committees.

Because the comparisons of OP and NP were not directly relevant to the description of our surveillance program, we removed this figure from our revised manuscript.

5. What sort of statistical analysis did you do to arrive at the statement "After careful consideration of the literature and further validation studies in our lab, we implemented a supervised self-collected anterior nares (AN) nasal

specimen." Are the "further validation studies" described anywhere?

To improve the clarity of the manuscript we deleted this rationale. Because the FDA treats NP and AN as equivalent upper respiratory samples, we opted for AN due to the ability to perform supervised self-collection. The only validation studies performed are described in the previous figure 2, which directly compared AN to NP swabs, and OP to NP swabs from symptomatic individuals. 

6. Who were the "healthy donors" in "We performed a comparator study using healthy donors to determine if sample integrity was maintained in the 1.4 mL tubes compared to the 15 mL sample collection tubes." What sorts of protocols were used for recruitment, informed consent, etc?

To improve clarity, this comparison is omitted in the final version of the manuscript. However, volunteers were recruited from the laboratory staff and patients visiting the UC Santa Cruz Student Health Services for COVID—19 tests. Participants provided informed consent for the study. 

7. Since there are already "blueprint" papers out there like this, it would be helpful to see how the methods outlined here compare to the previously published efforts. Currently, the manuscript reads almost like a methodology paper without comparison to existing methods.

We now provide detailed comparisons between the UC Berkeley IGI blueprint paper and our approach. This text is part of the discussion. We describe how our choice of sample collection tubes and liquid handlers enabled pooling and required a custom LIMS.

8. Please make sure that the claim "Although there are many reports of the theoretical implementation of sample pooling approaches for COVID-19 surveillance testing [33]; [34]; [35], there are few reports showing how this

process has been put into practice on an institutional level." is accurate. There are reports I'm aware of that describe this.

We revise the text according to the reviewers suggestion.

9. Is the LIMS you built truly the result of only a collaboration between your group and Third Wave Analytics? ("A custom laboratory information management system (LIMS) was developed in collaboration with Third Wave Analytics (San Francisco) using the Salesforce Lightning Platform and Experience Cloud.") Reading over the June 2020 paper from UC Berkeley, it looks like they also built such a system with Third Wave Analytics. If the system you built was at all an adaptation of this or a system another group built, it would be prudent to cite that group.

We followed UC Berkeley’s blueprint by contracting with Thirdwave, but our process (sample collection tubes, accessioning, liquid handling) was sufficiently different that developing a custom LIMS was necessary. This was particularly true for the pooled sample test. 

10. The section entitled "Impact of surveillance on campus-wide SARS-CoV2 transmission" is somewhat misleading. No epidemiological methods were employed in this paper to demonstrate that surveillance done affected overall transmission. We are not provided with any of the surveillance parameters for non-student groups (e.g., frequency, % of population of interest), nor prevalence in the local community.

We revised this section of the manuscript by highlighting a recently published epidemiology study by Pollock et al. that demonstrates that COVID—19 incidence in 19-29 year old demographic is significantly lower on UC Campuses than within the surrounding county. These data suggest that the multilayered mitigation strategies implemented by the UC, including asymptomatic surveillance, have an impact on SARS-CoV-2 transmission.

Pollock BH, Kilpatrick AM, Eisenman DP, Elton KL, Rutherford GW, Boden-Albala BM, et al. Safe reopening of college campuses during COVID-19: The University of California experience in Fall 2020. PLoS One. 2021;16: e0258738.

We also provide a comparison of the positivity rates for our campus surveillance plan and the clinical testing we performed for the safety-net health car providers (Figure 5D). 

11. Turn around time is mentioned in the introduction, but we are provided with no actual data about it in the body of the manuscript. Please provide analysis with the raw data.

We now include a plot and turnaround time statistics for all samples received by the diagnostic lab (Figure 2F).

12. Please check figure #s before resubmission.

We double checked all call outs to all figures in the revised manuscript

13. As a final note, the document has inconsistent formatting with paragraph breaks an indentation. It would be appreciated if revisions were properly formatted for ease of navigation.

We reformatted the manuscript prior to resubmission.

Reviewer #2: Overall this is a well written and clearly presented paper that does not require major revisions. It describes the practical implementation of a pooled testing strategy for automated handling, testing and tracking of a high volume of self-collected COVID-19 nasal swab samples, and should be of interest to audiences planning to implement or optimize strategies for high volume viral testing. While pooled strategies for campus COVID-19 surveillance have been widely mentioned in nontechnical news reports, this manuscript provides both technical details of implementation and useful supporting research that was conducted in order to validate the strategy and to optimize protocols.

1. In the results section, the statement "Surprisingly we observed a slight, but significant difference in Ct values for RP from negative individual surveillance samples and negative clinical samples (mean Ct 24.77 and 25.10, respectively)." remains as a bit of a loose end that needs to be explained more completely. The expectation (and observation) in the pooled surveillance samples was that there would be a difference in the two Ct values due to dilution -- why is there a different expectation for RP and what might be an explanation for the observed results?

This result is somewhat hard to explain. The individual surveillance samples refer to tests completed before we implemented pooling, so dilution is unlikely to be an explanation for this source of variation. Indeed, the pooled test only dilutes the SARS-CoV-2 signal, whereas RP is an endogenous gene and present at similar levels in all samples. The most logical explanation of this variability is in a clinician collected vs self-collected surveillance sample. We now raise this possibility in the revised manuscript. 

2. The abbreviation RP does not seem to be defined at its first occurrence.

We now define RP at the first occurrence in the revised manuscript.

3. Figure 1 might be less cluttered if presented as box and whisker plots, and I don't think the explicit connection between sample pairs is necessary to the point of the figure.

During the process of revising the manuscript, we decided to remove this figure because it distracts from the clarity of the paper. Instead, we replaced this figure with an overview of our processes.

4. Figure 2 labels contain misspellings of the word "Individual".

We corrected this typo in Figure 2. We also improved the labeling by referring to pooled or unpooled samples.

5. Are automation programs for the robotics and custom LIMS code available anywhere e.g. on a GitHub? It seems like those might be useful for other teams trying to implement similar protocols or using the same hardware for related purposes.

We created a github account and uploaded all automation scripts. We refer to this repository in the methods section. The github account is available here:

https://github.com/UCSC-CCDL/Bravo-protocol-files

---

## [Editor Report · Decision Letter 1]

25 Nov 2021

A method for campus-wide SARS-CoV-2 surveillance at a large public university

PONE-D-21-23418R1

Dear Dr. Sanford,

We’re pleased to inform you that your manuscript has been judged scientifically suitable for publication and will be formally accepted for publication once it meets all outstanding technical requirements.

Kind regards,

David M. Ojcius

Academic Editor

PLOS ONE
---

## [Editor Report · Acceptance letter]

9 Dec 2021

PONE-D-21-23418R1 

A method for campus-wide SARS-CoV-2 surveillance at a large public university 

Dear Dr. Sanford:

I'm pleased to inform you that your manuscript has been deemed suitable for publication in PLOS ONE. Congratulations! Your manuscript is now with our production department. 

Kind regards, 

on behalf of

Dr. David M. Ojcius 

Academic Editor

PLOS ONE